# NeuroSymAD: A Neuro-Symbolic Framework for Interpretable Alzheimer's Disease Diagnosis

## Abstract

Early diagnosis of Alzheimer's disease (AD) by integrating neuroimaging and clinical data shows great potential for accurate assessment. While deep learning technique achieves tremendous success, it often functions as a black box, limiting interpretability and lacking mechanisms to effectively integrate critical clinical data such as biomarkers, medical history, and demographic information. To bridge this gap, we propose NeuroSymAD, a neuro-symbolic framework that synergizes neural networks with symbolic reasoning. A neural network perceives brain MRI scans, while a large language model (LLM) distills medical rules to guide a symbolic system in reasoning over biomarkers and medical history. This structured integration enhances both diagnostic accuracy and explainability. Experiments on the ADNI dataset demonstrate that NeuroSymAD outperforms state-of-the-art methods by up to 2.91% in accuracy and 3.43% in F1-score while providing transparent and interpretable diagnosis.

## 1 Introduction

Alzheimer's disease (AD) Blennow et al. (2006); Scheltens et al. (2016) is one of the most pressing healthcare challenges in the aging society, affecting millions globally—a number projected to triple by 2050. This surge not only threatens public health but also imposes profound socioeconomic burdens. Accurate diagnosis of AD is crucial for timely intervention and better patient outcomes DeTure & Dickson (2019). Currently, AD diagnosis relies heavily on the comprehensive analysis of multiple data modalities, including brain MRI scans Jack Jr et al. (2008), various biomarkers Jack & Holtzman (2013), and patient history Bateman et al. (2012). Clinicians follow a sophisticated diagnostic process that integrates visual image analysis with reasoning based on extensive medical knowledge and expertise. This process exemplifies a unique integration of perceptual skills in image analysis and logical reasoning with domain expertise, effective yet challenging to automate.

Recent advances in deep learning have proven effective for AD diagnosis using neuroimaging. For instance, 3D CNNs and 3D ResNets have achieved high accuracy in classifying AD from MRI scans Mirzaei et al. (2016); Alsubaie et al. (2024a); AbdulAzeem et al. (2021); Al Shehri (2022); Turrisi et al. (2023); Ebrahimi et al. (2020); Farooq et al. (2017); Mohi ud din dar et al. (2023). However, such models typically operate as "black boxes" and are limited by single-modality inputs. To address this, multimodal frameworks integrating imaging, genetic, and clinical data have been developed Qiu et al. (2022); Golovanevsky et al. (2022); Liu et al. (2018); Elazab et al. (2024); Zhang et al. (2023b); Venkatraman et al. (2024). Some works have also innovated on model architectures to enhance performance Zhang et al. (2023b); Venkatraman et al. (2024); Alsubaie et al. (2024b); Choudhury et al. (2024); Yaqoob et al. (2024); Shaffi et al. (2024). While these multimodal approaches improve diagnostic coverage, they continue to function as opaque end-to-end models that offer little transparency into how individual clinical factors contribute to a diagnosis, which is a critical requirement for clinical trust and regulatory acceptance. Furthermore, most architectures still lack a principled mechanism to incorporate structured medical knowledge such as clinical guidelines or established risk factor relationships, leaving a gap between model predictions and clinician reasoning.

In contrast, symbolic systems can explicitly encode medical knowledge but struggle with handling visual data and require substantial manual effort in rule construction and updating. Neuro-symbolic methods have recently been explored to bridge this gap. For example, PP-DKL Lavin (2021) employs probabilistic programming to predict early neurodegeneration. However, it relies on single-modality data and does not fully integrate clinical information. These limitations motivate the development of a neuro-symbolic system that merges deep learning's perceptual strengths with the interpretability of symbolic reasoning, mimicking clinicians' diagnostic process.

In this paper, we propose *NeuroSymAD*, a neuro-symbolic AD diagnosis framework that seamlessly integrates deep learning–based image perception with knowledge-driven symbolic reasoning. Our contributions are: (1) a unified framework that mimics clinical experts' diagnostic process by combining neural perception with symbolic reasoning; (2) an automated knowledge acquisition module that constructs and updates the rule base efficiently; (3) an end-to-end training strategy that jointly optimizes neural and symbolic components, allowing medical knowledge to guide representation learning while adapting rules to data; and (4) extensive evaluation on clinical datasets, demonstrating superior diagnostic accuracy and interpretability over state-of-the-art baselines.

## 2 Related Work

**Unimodal deep learning for AD diagnosis.** Early deep learning approaches to AD diagnosis focused primarily on neuroimaging. Architectures such as 3D CNNs and 3D ResNets have demonstrated strong classification performance on MRI scans (Mirzaei et al., 2016; Alsubaie et al., 2024a; AbdulAzeem et al., 2021; Al Shehri, 2022; Turrisi et al., 2023; Ebrahimi et al., 2020; Farooq et al., 2017; Mohi ud din dar et al., 2023). Subsequent architectural innovations, including dual-attention mechanisms (Venkatraman et al., 2024), novel CNN designs (Alsubaie et al., 2024b), coupled GAN-based fusion (Choudhury et al., 2024), Bayesian-optimized ResNets (Yaqoob et al., 2024), and ensemble vision transformers (Shaffi et al., 2024), have further improved accuracy. Despite these advances, unimodal models are inherently constrained by their reliance on imaging alone and offer little transparency into their predictions, limiting their suitability for clinical deployment.

**Multimodal approaches.** To overcome the limitations of single-modality inputs, multimodal frameworks have been proposed that integrate MRI with genetic markers, PET scans, and structured clinical variables (Qiu et al., 2022; Golovanevsky et al., 2022; Liu et al., 2018; Elazab et al., 2024; Zhang et al., 2023a). Notable examples include 3MT (Liu et al., 2023), a cascaded multi-modal mixing Transformer that incorporates MRI with demographic and clinical variables, and MM-GNN (Zhang et al., 2023a), which fuses MRI, PET, and clinical attributes via graph neural networks. While these methods broaden the input space and often achieve high reported accuracy, they frequently rely on variables such as MMSE scores or PET imaging that are not routinely available in clinical practice, and they remain fundamentally opaque in their decision-making. Critically, none of these approaches provides a principled mechanism for incorporating structured medical knowledge or generating rule-based explanations that clinicians can audit and trust.

**Neuro-symbolic methods.** Neuro-symbolic approaches seek to combine the perceptual capacity of neural networks with the interpretability of symbolic reasoning. In the context of neurodegeneration, PP-DKL (Lavin, 2021) employs probabilistic programming over deep kernels to model early disease progression. However, PP-DKL operates on single-modality data and does not integrate the demographic, clinical, or medical history information that clinicians rely upon. More broadly, existing neuro-symbolic systems in medical AI either require extensive manual rule curation or lack end-to-end optimization, preventing neural and symbolic components from mutually reinforcing each other during training. NeuroSymAD addresses these gaps by integrating MRI perception with an automatically constructed symbolic rule base, jointly optimizing both components, and generating transparent diagnostic reports grounded in clinical knowledge.

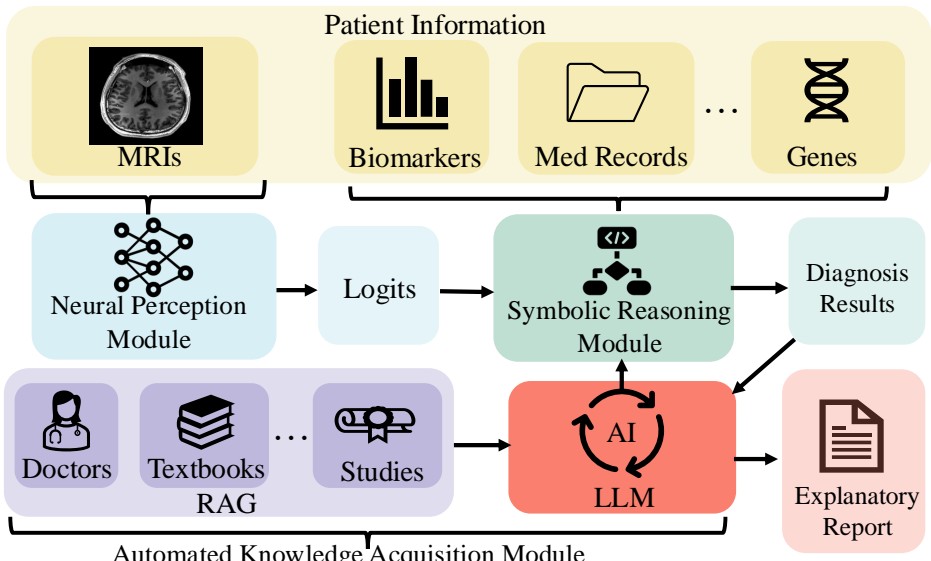

Figure 1: NeuroSymAD architecture. Inputs comprise a 3D MRI and structured patient attributes. The neural perception module encodes MRI into logits for CN vs. AD. The symbolic reasoning module applies a learned rule set, generated by an LLM with retrieval-augmented knowledge, to compute a differentiable adjustment, producing adjusted logits. An MRI-only pretraining stage followed by end-to-end joint optimization aligns neural and symbolic components. The knowledge module also generates an explanatory report summarizing activated rules and contributing factors to support clinical decision-making.

## 3 Methods

### 3.1 Overview

Our AD diagnosis neuro-symbolic framework comprises three key components: a neural perception module, a symbolic reasoning module, and an automated knowledge acquisition module, as illustrated in Figure 1. Given a patient sample, including a 3D MRI scan $\mathbf{x}_i \in \mathbb{R}^{m \times n \times r}$ and a set of patient's information and medical history records $\mathbf{z}_i \in \mathbb{R}^k$, the neural perception module employs a deep neural network to process the MRI scan and estimate the probability of AD. This estimation is represented as logits $\mathbf{y_i} \in \mathbb{R}^2$, corresponding to two classes: CN and AD. Next, the symbolic reasoning module refines these logits based on the patient information $\mathbf{z}_i$ by leveraging a set of learned rules to enhance diagnostic accuracy. These rules are generated by the automated knowledge acquisition module, which is composed of a Large Language Model (LLM) integrated with a Retrieval-Augmented Generation (RAG) system. Furthermore, an end-to-end training strategy is employed to jointly optimize the neural perception module and symbolic reasoning module. Finally, the automated knowledge acquisition module consolidates the adjusted logits and reasoning rules using the LLM to generate an explanatory report to show the reasoning process, supporting clinical decision-making.

### 3.2 Neural Perception Module

The neural perception module processes MRI scans using a deep learning model, denoted as $\mathcal{F}$, mapping an MRI scan to probability logits:

$$\mathcal{F} : \mathbb{R}^{m \times n \times r} \to \mathbb{R}^2, \quad \mathbf{y_i} = \mathcal{F}(\mathbf{x_i}), \tag{1}$$

where $\mathbf{y_i}$ represents the predicted probability logits for the two classes. To optimize the model, we first pretrain it solely on MRI scans using a cross-entropy loss function, optimized via an Adam optimizer(Adam

et al., 2014):

$$\mathcal{L} = -\frac{1}{N} \sum_{i=1}^{N} \left[ \hat{y}_i \log y_i + (1 - \hat{y}_i) \log(1 - y_i) \right], \tag{2}$$

where $\hat{y}_i$ is the ground-truth diagnosis label for $\mathbf{x}_i$, and $y_i$, obtained as $y_i = \text{softmax}(\mathbf{y}_i)$, is the predicted label. After this MRI-only pretraining, the model is further refined through end-to-end training with the symbolic reasoning module.

### 3.3 Symbolic Reasoning Module

The symbolic reasoning module enhances the neural network's predictions by integrating domain knowledge in the form of symbolic rules. An LLM equipped with an AD clinical knowledge RAG system generates these rules, capturing the relationships between demographic, clinical characteristics, medical history records, and classification logits. Given the initial logits $\mathbf{y}_i$ from the neural network, we apply a set of symbolic rules $\mathcal{R} = \{R_1, R_2, \ldots, R_K\}$ generated by the LLM. The symbolic reasoning process first analyzes the patient information, biomarkers and medical history records $\mathbf{z}_i$ to determine which subset of rules $\mathcal{R}_i \subseteq \mathcal{R}$ are relevant for the current patient. Specifically, a rule $\mathcal{R}_j$ is activated for patient $i$ if the corresponding clinical variable satisfies a predefined condition, such as falling within a threshold range or exceeding a critical value. The activated rules are then applied through a set of differentiable operations to compute adjustments to the logits.

For each rule, we define specific differentiable operations with learnable parameters that capture various medical relationships. The age-related rule provides an example of how medical knowledge is encoded in our framework:

$$\delta_{\text{age}} = \alpha \cdot \sigma \left( \frac{z_{\text{age}} - T_1}{\tau} \right) + \beta \cdot \text{ReLU}(z_{\text{age}} - T_2), \tag{3}$$

where $\alpha, \beta, T_1, T_2$ are trainable parameters. $\alpha$ is the base effect strength, $T_1$ is the age threshold, $\beta$ is the acceleration factor, and $T_2$ is the acceleration threshold. $\tau$ controls the smoothness of the sigmoid transition $\sigma(\cdot)$, which models a smooth transition in risk at the threshold age, while the second term captures accelerated risk increase in advanced age. This formulation encodes the clinical knowledge that AD risk increases after a certain age and accelerates further in later years Isik (2010). By training these parameters, the model automatically determines at what ages these effects become significant and how strong each effect is.

Similar differentiable formulations are defined for other rules, including gender-specific factors, education level, comorbidities, lifestyle factors, and clinical indicators. Let $\delta_i = \sum_{j \in \mathcal{R}_i} \delta_{i,j}$ aggregate the effects of all relevant rules. The final adjusted logits are obtained by applying this cumulative adjustment:

$$\tilde{\mathbf{y}}_i = \mathbf{y}_i + [-w \cdot \delta_i, \ \delta_i]^T, \tag{4}$$

where $w$ is a balance factor. The flexible formulations allow us to incorporate complex domain-specific relationships in a differentiable manner, enabling end-to-end training. The learnable parameters of each rule operation are optimized during training, allowing the model to automatically determine the appropriate effect of each medical factor based on empirical data while maintaining the structure informed by clinical knowledge.

After pretraining the neural perception module on MRIs alone, we perform end-to-end training by incorporating patients' information and medical history. In this phase, both the neural network weights and the trainable rule parameters in the symbolic system are jointly optimized. This not only allows the symbolic rules to align better with empirical data but also leverages medical knowledge to guide the optimization of the neural network. This bidirectional interaction between neural and symbolic components enables NeuroSymAD to achieve high accuracy and interpretability, making it more suitable for clinical applications.

### 3.4 Automated Knowledge Acquisition Module

The knowledge acquisition module is designed to bridge the gap between unstructured medical knowledge and symbolic reasoning within our framework. It performs two major functions: automatic rule generation

and explanatory report construction. For rule generation, the module leverages an LLM integrated with a RAG system to systematically extract symbolic rules from diverse clinical sources, including diagnostic guidelines, peer-reviewed literature, and standard medical textbooks. The pipeline consists of three stages: (1) retrieval of relevant domain documents guided by clinical queries, (2) extraction of key medical insights such as risk modifiers, protective factors, and biomarker thresholds, and (3) transformation of these insights into formalized logical statements that explicitly capture relationships between demographic variables, biomarkers, medical history, and diagnostic outcomes. The extracted knowledge is converted into formulations with trainable parameters in the neuro-symbolic system during end-to-end training. Beyond rule generation, the module also ensures scalability and robustness of the symbolic knowledge base. Because new literature and clinical guidelines emerge continuously, the LLM-RAG pipeline supports dynamic updating, allowing the rule set to expand and remain aligned with the evolving medical consensus. This capability substantially reduces manual curation effort and ensures that the system reflects state-of-the-art domain knowledge.

During inference, the module generates explanatory reports by combining both the neural network output and the adjustments introduced by the neuro-symbolic reasoning system. Specifically, it first presents the raw probability of AD predicted from brain MRI scans by the neural backbone, and then explains how symbolic rules modify this estimate by incorporating clinical or demographic factors. The final report thus makes explicit which rules were activated, how they influenced the prediction, and what the adjusted diagnostic conclusion is. This dual-perspective explanation not only clarifies the interaction between neural and symbolic components but also enhances transparency and clinical trust.

### 3.5 Training and Implementation Details

We adopt a two-stage training strategy. In stage one, the neural perception module is trained on MRI-only inputs to learn robust perceptual features. In stage two, we incorporate the symbolic reasoning module and fine-tune the entire system end-to-end with a lower learning rate. MRI scans are resampled to $128 \times 128 \times 128$ resolution; standard normalization and data augmentation (random rotations and flips) are applied. We use weighted cross-entropy loss to handle class imbalance. Optimization uses Adam with initial learning rates of $10^{-4}$ (stage one) and $10^{-5}$ (stage two), a step scheduler ($\gamma = 0.5$, step size $= 10$), batch size $= 8$, and 30 epochs per stage.

## 4 Experiments

To ground the study and clarify the experimental setting from the outset, we evaluate NeuroSymAD on the Alzheimer's Disease Neuroimaging Initiative (ADNI1–3) cohort Mueller et al. (2005), using T1-weighted MRI from 3,088 individuals. Standard preprocessing includes skull stripping via FreeSurfer `recon-all` Fischl (2012). Following common practice and to mitigate class imbalance while reflecting disease severity, we merge CN/SMC/EMCI into a single cognitively normal ("CN") group and LMCI/AD into "AD" Jack Jr et al. (2018). For comparison, we consider strong unimodal CNN baselines, including DenseNet-169 Al Shehri (2022), an optimized 3D CNN tailored for AD MRI Turrisi et al. (2023), a Dual-Attention CNN Venkatraman et al. (2024), and 3D ResNet Ebrahimi et al. (2020). We also evaluate representative multimodal methods, including 3MT (a cascaded multi-modal mixing Transformer integrating MRI with demographic/clinical variables) Liu et al. (2023), MM-GNN (a graph-based fusion of MRI, PET, and clinical attributes) Zhang et al. (2023a).

To ensure a fair and clinically realistic comparison, we intentionally *exclude* cognitive test scores such as MMSE, additional imaging such as PET, and fluid or blood biomarkers that are tightly coupled to AD pathology and clinical labels. In particular, we remove $A\beta$ and tau measurements and their ratios. These variables either encode diagnostic decisions directly or are not routinely available in practice. When reproducing multimodal baselines, we therefore align their inputs to the same MRI plus basic demographic/clinical setting used by NeuroSymAD. Under this protocol, our primary task is binary AD versus CN classification with the above grouping; evaluation focuses on accuracy, precision, recall, F1-score, and AUC, complemented by ablation analyses across backbones and qualitative assessments of case-level explanations and attention visualizations. This design allows us to isolate the contribution of neuro-symbolic reasoning from that of strong diagnostic variables, while keeping the study consistent with realistic clinical workflows.

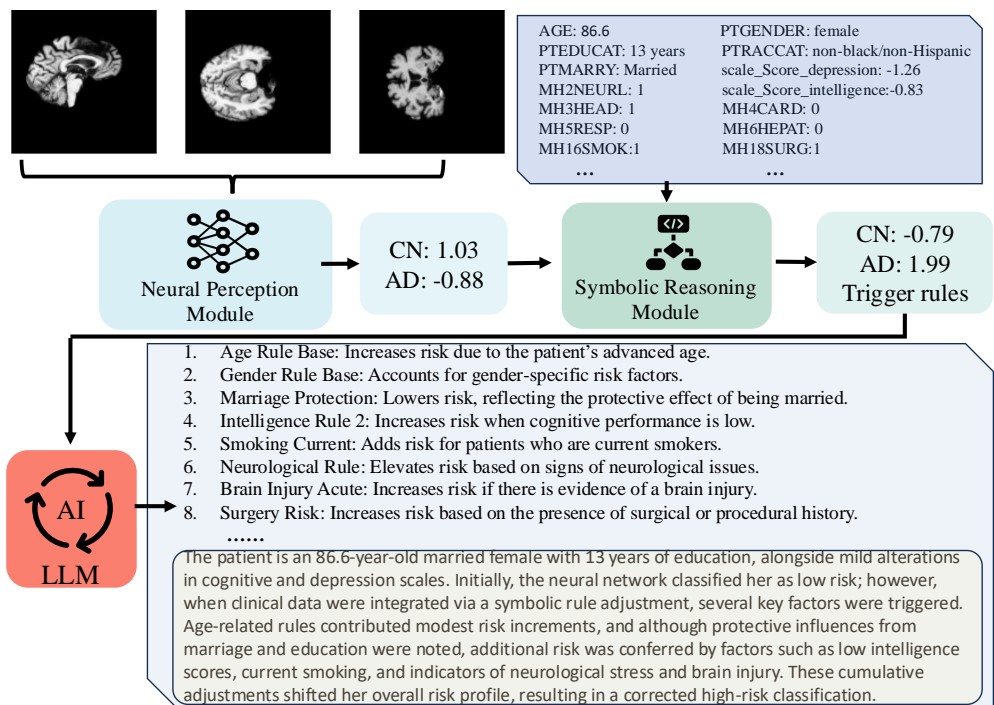

Figure 2: An example of NeuroSymAD's diagnostic process and the generated explanatory report, demonstrating how the symbolic reasoning module corrects the misclassification of neural network.

***Overall performance.*** Table 1 presents the performance comparison of NeuroSymAD against unimodal and multimodal baselines. NeuroSymAD outperforms all unimodal CNN baselines across key metrics, achieving 88.58% accuracy and the highest F1-score (92.15%). Although multimodal methods such as 3MT have reported very high performance in the literature, often above 95%, they generally rely on MMSE or PET, modalities that either encode diagnosis directly (MMSE) or are not available routinely in practice.

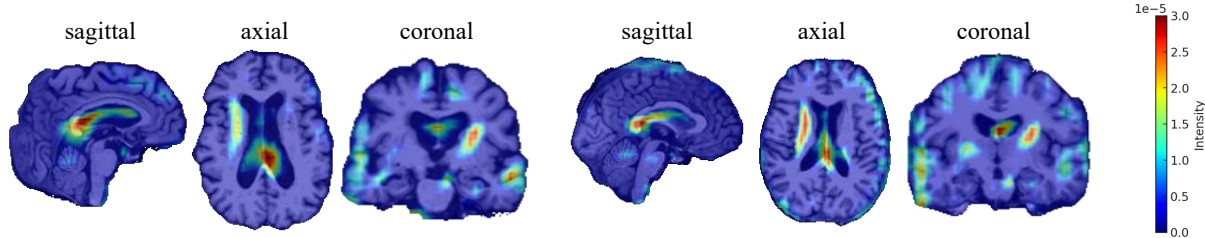

Figure 3: Heatmaps overlaid on MRIs for two AD subjects, showing the activation distribution of NeuroSymAD.

***Ablation study.*** Table 2 shows that symbolic reasoning consistently improves all CNN backbones. Gains in precision and F1 are statistically significant ($p < 0.05$), confirming the robustness of symbolic integration.

***Case study and visualization.*** Figure 2 illustrates how symbolic reasoning corrected a neural misclassification by integrating patient-specific risk factors. Figure 3 highlights clinically relevant regions activated by NeuroSymAD, aligning with known neuropathological findings. Table 3 summarizes the rules and hypotheses generated by the LLM enhanced with the RAG system. Figure 4 presents the corresponding marginal contributions of these symbolic rules to the diagnostic performance.

Table 1: Performance comparison of NeuroSymAD against unimodal and multimodal methods for AD diagnosis (mean ± standard deviation over 10 random seeds). *Row shading indicates modality:* light blue = unimodal, light orange = multimodal, light green = **Ours**. Best results are in **bold**.

| Method | Accuracy | Precision | Recall | F1 | AUC |
|---|---|---|---|---|---|
| DenseNet-169 Al Shehri (2022) | $81.64_{\pm1.84}$ | $82.40_{\pm0.81}$ | $79.21_{\pm1.32}$ | $80.78_{\pm0.78}$ | $81.54_{\pm1.32}$ |
| Opt 3D CNN Turrisi et al. (2023) | $83.30_{\pm1.52}$ | $80.21_{\pm2.02}$ | $92.42_{\pm1.78}$ | $85.88_{\pm0.89}$ | $89.20_{\pm1.24}$ |
| DA CNN Venkatraman et al. (2024) | $86.42_{\pm1.32}$ | $86.88_{\pm1.24}$ | $93.24_{\pm1.45}$ | $88.46_{\pm0.54}$ | $89.21_{\pm0.87}$ |
| 3D ResNet Ebrahimi et al. (2020) | $85.67_{\pm1.20}$ | $86.16_{\pm3.08}$ | $\mathbf{95.25}_{\pm2.37}$ | $90.46_{\pm0.64}$ | $\mathbf{93.36}_{\pm0.47}$ |
| 3MT Liu et al. (2023) | $86.34_{\pm1.2}$ | $85.50_{\pm1.3}$ | $90.30_{\pm1.5}$ | $87.72_{\pm1.0}$ | $88.10_{\pm1.1}$ |
| MM-GNN Zhang et al. (2023a) | $85.82_{\pm1.5}$ | $85.21_{\pm1.4}$ | $89.50_{\pm1.6}$ | $87.34_{\pm1.2}$ | $87.92_{\pm1.3}$ |
| GPT-4o Hurst et al. (2024) | $36.02_{\pm1.8}$ | $35.12_{\pm2.5}$ | $28.32_{\pm2.3}$ | $31.33_{\pm1.8}$ | $44.10_{\pm1.1}$ |
| **NeuroSymAD (Ours)** | $\mathbf{88.58}_{\pm1.75}$ | $\mathbf{89.97}_{\pm0.83}$ | $94.44_{\pm1.69}$ | $\mathbf{92.15}_{\pm0.93}$ | $92.56_{\pm1.59}$ |

Table 2: Comparison of our method with existing neural networks. Starred metrics (*) indicate a significant improvement ($p < 0.05$, paired t-test) with symbolic components.

| Model | Method | Accuracy | Precision | Recall | F1 | AUC |
|---|---|---|---|---|---|---|
| DenseNet-169 Al Shehri (2022) | Base | $81.64_{\pm1.84}$ | $82.40_{\pm0.81}$ | $79.21_{\pm1.32}$ | $80.78_{\pm0.78}$ | $81.54_{\pm1.32}$ |
| | Ours | $\mathbf{83.63}^{*}_{\pm0.84}$ | $\mathbf{85.24}^{*}_{\pm1.24}$ | $\mathbf{83.20}^{*}_{\pm0.90}$ | $\mathbf{84.21}^{*}_{\pm0.98}$ | $\mathbf{82.31}_{\pm1.42}$ |
| Opt 3D CNN Turrisi et al. (2023) | Base | $83.30_{\pm1.52}$ | $80.21_{\pm2.02}$ | $92.42_{\pm1.78}$ | $85.88_{\pm0.89}$ | $89.20_{\pm1.24}$ |
| | Ours | $\mathbf{85.81}^{*}_{\pm2.13}$ | $\mathbf{83.45}^{*}_{\pm1.86}$ | $\mathbf{93.05}^{*}_{\pm2.03}$ | $\mathbf{87.98}^{*}_{\pm1.92}$ | $\mathbf{90.10}^{*}_{\pm0.88}$ |
| DA CNN Venkatraman et al. (2024) | Base | $86.42_{\pm1.32}$ | $86.88_{\pm1.24}$ | $\mathbf{93.24}_{\pm1.45}$ | $88.46_{\pm0.54}$ | $89.21_{\pm0.87}$ |
| | Ours | $\mathbf{87.32}_{\pm1.12}$ | $\mathbf{88.35}^{*}_{\pm2.32}$ | $91.23^{*}_{\pm1.65}$ | $\mathbf{89.76}^{*}_{\pm1.40}$ | $89.40_{\pm0.35}$ |
| 3D ResNet Ebrahimi et al. (2020) | Base | $85.67_{\pm1.20}$ | $86.16_{\pm3.08}$ | $\mathbf{95.25}_{\pm2.37}$ | $90.46_{\pm0.64}$ | $\mathbf{93.36}_{\pm0.47}$ |
| | Ours | $\mathbf{88.58}^{*}_{\pm1.75}$ | $\mathbf{89.97}^{*}_{\pm0.83}$ | $94.44_{\pm1.69}$ | $\mathbf{92.15}^{*}_{\pm0.93}$ | $92.56_{\pm1.59}$ |

## 5 Discussion

NeuroSymAD consistently improves over unimodal CNN baselines in accuracy, precision, and F1-score, while also generating interpretable diagnostic reports through symbolic rules. This interpretability addresses a key limitation of black-box deep learning in clinical practice.

Multimodal baselines such as 3MT Liu et al. (2023), MM-GNN Zhang et al. (2023a) report high accuracies, but these results largely rely on MMSE (which nearly encodes diagnosis), PET imaging, or fluid/blood biomarkers (which are not routinely available in clinical settings). Such variables are rarely accessible at the actual diagnostic stage and may artificially inflate performance, making it difficult to assess the real contribution of the model architecture. In contrast, NeuroSymAD uses only the imaging and information that clinicians would realistically have access to, such as MRI and basic demographic or clinical risk factors. Under these realistic clinical inputs, our approach consistently outperforms the baselines, demonstrating that it is both more practical and more effective for real-world diagnostic scenarios.

Our case study illustrates how symbolic reasoning can correct neural misclassifications by incorporating patient-specific risk factors, mirroring expert diagnostic logic. Heatmap analyses further align with clinically relevant brain regions, reinforcing the plausibility of the learned representations. Limitations include reliance on ADNI and coverage of symbolic knowledge. Future directions involve external validation, longitudinal modeling, and expansion of symbolic rules guided by clinical practice.

In order to quantify the contribution of different symbolic rules, we evaluate their group-wise marginal effects on a backbone 3D-ResNet model. The results are summarized in Figure 4. The left panel reports the improvement in accuracy (ΔAccuracy), while the right panel shows the reduction in false positive rates (ΔSpecificity). We observe that certain symbolic rules, such as *education* and *marital status*, play a significant role. For example, *education* increases accuracy by approximately 0.9% and reduces false positives by more than 5.5%, whereas *marital status* improves accuracy by about 0.2% and specificity by 1.8%, which aligns with clinical findings Stern (2012); Sundström et al. (2014); Meng & D'arcy (2012). In contrast, factors such

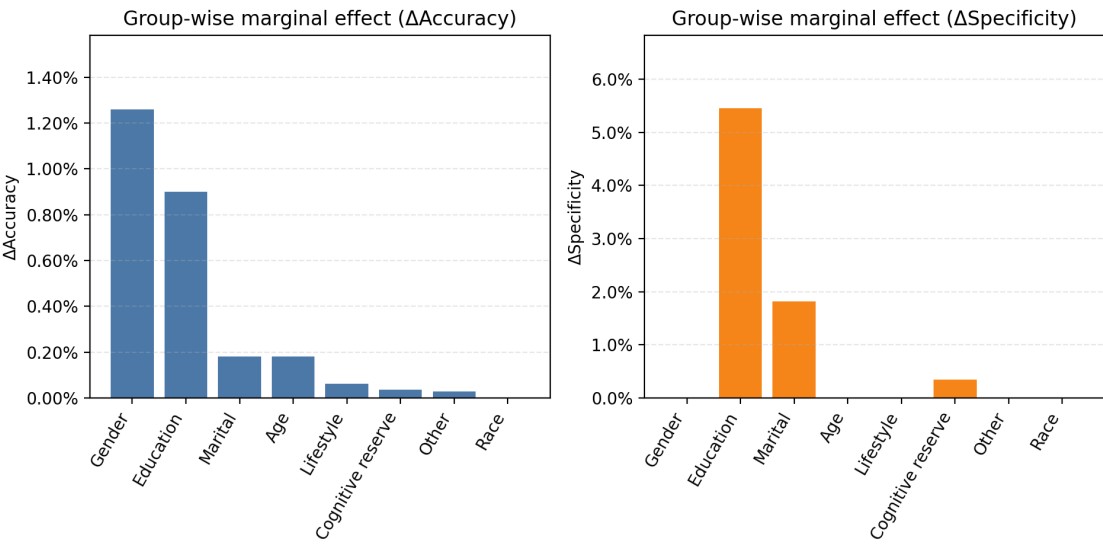

Figure 4: Grouped neuro-symbolic ablation. Bars show the marginal performance improvements attributable to each rule group on the test set. Left: ΔAccuracy, right: ΔSpecificity (higher is better; effects are non-additive).

Table 3: Symbolic rules and hypotheses generated by an LLM with RAG.

| Group | Example effects / assumptions |
| --- | --- |
| **Gender** | Sex-related adjustment for female participants
Post-menopausal increase in risk after typical menopause age
Midlife age-window interaction modulating female risk |
| **Education** | Protection above an education threshold (secondary/tertiary completion)
Linear protection per additional year of schooling (cognitive reserve)
Education–age interaction with stronger reserve in later life |
| **Marital** | Social/partner-support protection for married participants |
| **Age** | Late-life baseline risk increase with advancing age
Additional risk beyond a higher critical age threshold
Accelerated risk growth at very old ages |
| **Race** | Risk modulation for Black participants
Risk modulation for Hispanic participants |
| **Cognitive reserve** | Protection proportional to higher cognitive test scores
Additional reserve when scores exceed a threshold |
| **Lifestyle** | Increased risk due to current smoking
Residual risk associated with past smoking
Potential protective/neutral effect of moderate alcohol use
Increased risk due to heavy alcohol consumption |
| **Other** | Other clinical and systemic modifiers, such as cardiorespiratory comorbidities, endocrine/metabolic indicators, sleep and depression proxies, prior head injury or loss-of-consciousness, and a small global calibration term, whose effects are generally sparser and weaker than the main groups |

as *race* and *lifestyle* contribute negligibly, with changes close to zero in both metrics. We further note that the relatively large improvement of *gender* on accuracy is consistent with clinical observations that women are more susceptible to Alzheimer's disease Vina & Lloret (2010), and our symbolic system provides supporting evidence for this effect. These findings indicate that our symbolic system can automatically distinguish useful symbolic rules from irrelevant ones, thereby providing interpretability. Moreover, the analysis reveals not only the existence of impactful factors but also their relative importance. For instance, the effect of *education* consistently surpasses that of *marital status*, suggesting that the system can shed new light on the weight of symbolic rules.

Overall, NeuroSymAD strikes a balance between predictive performance, interpretability, and practicality, offering a complementary path to multimodal deep learning approaches while maintaining fairness and transparency.

## 6 Conclusion

NeuroSymAD presents a neuro-symbolic framework for Alzheimer's disease diagnosis that integrates a 3D-ResNet backbone with symbolic reasoning rules distilled by an LLM with retrieval-augmented knowledge. On ADNI with clinically realistic inputs, the framework improves accuracy, precision, and F1 over strong CNN baselines while providing transparent, rule-based explanations. Ablations highlight the contributions of factors such as education, marital status, and gender, aligning with established clinical observations. These findings show that NeuroSymAD enhances predictive performance and interpretability simultaneously, offering a practical and extensible paradigm for trustworthy diagnostic support in neurodegenerative disorders.

### Broader Impact Statement

NeuroSymAD advances the development of trustworthy clinical AI by combining predictive accuracy with transparent, rule-based reasoning. By making diagnostic logic explicitly auditable, the framework supports rather than supplants clinical judgment, offering a practical tool for early Alzheimer's detection that clinicians can interrogate and trust. More broadly, the neuro-symbolic paradigm demonstrated here may inform interpretable AI development across other neurodegenerative and chronic disease domains.

Nonetheless, two risk factors warrant consideration. First, as the model is trained and validated exclusively on ADNI, a dataset that skews toward well-resourced, predominantly White North American cohorts, its diagnostic rules and learned thresholds may not generalize equitably across diverse ethnicities, healthcare systems, or imaging protocols. Deployment without external validation on representative populations risks reinforcing existing health disparities. Second, the framework processes sensitive patient information including demographic attributes, medical history, and neuroimaging data. Any clinical deployment must adhere to applicable data protection regulations and ensure that patient records are handled with appropriate access controls and de-identification protocols.

We therefore advocate that NeuroSymAD be positioned as a decision-support tool subject to clinician oversight, and that future work prioritize external validation across diverse cohorts before any real-world deployment is considered.

### Acknowledgments

We thank the Alzheimer's Disease Neuroimaging Initiative (ADNI) investigators and coordinators for providing access to the dataset used in this study. Data were obtained from ADNI `https://adni.loni.usc.edu`; ADNI investigators contributed to the design and implementation of ADNI but did not participate in the analysis or writing of this paper.

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
