# OpenReview forum: "NeuroSymAD: A Neuro-Symbolic Framework for Interpretable Alzheimer's Disease Diagnosis"
_TMLR — Withdrawn by Authors_

### Review · Reviewer_4SCR · 2026-04-30

**Summary Of Contributions:**

The paper proposes NeuroSymAD, a framework for binary Alzheimer's vs. cognitively-normal classification that pairs a 3D-CNN over MRI with a tabular model operating on demographic and medical-history variables. The MRI backbone produces CN/AD logits, and a second model adds a learned scalar correction computed as a sum of per-variable parametric shape functions (sigmoid + ReLU with trainable thresholds and slopes). The variables to include and the shape-function templates to apply are selected offline by an LLM with RAG access to clinical literature; the same LLM also generates post-hoc natural-language explanatory reports. Training is two-stage: MRI-only pretraining of the CNN, followed by joint fine-tuning of CNN weights and the parametric rule parameters under a single cross-entropy loss on the summed logits. The authors evaluate on ADNI1–3 under a "fair-input" protocol that deliberately excludes MMSE, PET, and fluid biomarkers on the grounds that these variables either encode the diagnostic label or are not routinely available at the time of diagnosis. Under this protocol, NeuroSymAD reports 88.58% accuracy and 92.15% F1, outperforming unimodal CNN baselines by roughly 2–3 points and outperforming re-implemented multimodal baselines (3MT, MM-GNN) under matched inputs. A group-wise ablation attributes most of the gain to education, gender, and marital status, with race and lifetime alcohol use contributing negligibly.

**Audience:**

Yes

**Audience Explanation:**

The medical-imaging-ML community working on Alzheimer's and on combining neuroimaging with structured clinical variables would benefit from this paper — primarily for the fair-input evaluation protocol and the re-benchmarking of multimodal baselines under that protocol, which addresses a recurring label-leakage problem in this literature. Researchers interested in interpretable clinical decision support would also find the rule-level marginal-effect analysis and patient-level explanatory reports a useful design reference. However, the neuro-symbolic AI and interpretable-ML communities are less likely to find novel methodology here.

**Claims And Evidence:**

No

**Claims Explanation:**

1. The "neuro-symbolic" claim is not supported by the methods. Equations 3–4 implement a generalized additive model with smooth parametric shape functions added to CNN logits. There is no logical formalism, no rule composition, no discrete reasoning beyond per-variable threshold gates, and no probabilistic logic. The paper provides no evidence that what it calls "symbolic reasoning" is distinct from a parametric tabular head. The label is a framing claim, not an empirical one, and as written it is unsupported.

2. No baselines isolate the contribution of the rule structure. The central claim — that LLM-distilled symbolic rules improve diagnosis — requires comparison against simpler tabular-fusion alternatives that share the same input variables: CNN logits + logistic regression, GAM/EBM, or gradient-boosted trees on the demographic vector. None are reported. Without these, the gains over unimodal CNNs cannot be attributed to the symbolic apparatus rather than to generic tabular fusion.

3. The LLM's contribution is not isolated. The LLM-RAG pipeline is sold as a key contribution (automated knowledge acquisition), but the LLM is offline and not in the trained model's inference path. A hand-curated rule set would test whether the LLM adds anything beyond standard clinician-driven feature engineering. This ablation is absent.

4. The GPT-4o baseline at 36% accuracy is anomalous and unexplained. A general-purpose LLM performing below chance-adjacent on a binary task warrants methodological detail (input format, prompt, whether MRI was provided) that the paper does not give. As reported, this row is not interpretable evidence.

**Requested Changes:**

1. Add the missing tabular-fusion baselines. The central empirical claim — that LLM-distilled symbolic rules improve diagnosis — cannot be evaluated without comparison against simpler tabular-fusion alternatives that share the same input variables. Please add, under the same fair-input protocol and the same 3D-ResNet backbone:

- CNN logits + logistic regression on the tabular vector $z_i$ (linear shape functions).

- CNN logits + EBM / GA²M (data-driven smooth shape functions, the standard interpretable-GAM baseline).

- CNN logits + gradient-boosted trees (e.g., XGBoost / LightGBM) to capture interactions.

2. Reframe the "neuro-symbolic" claim. Eq. 3–4 implement a generalized additive model with smooth parametric shape functions added to CNN logits. I did not find any logical formalism, no rule composition, no probabilistic logic, and no discrete reasoning beyond per-variable threshold gates. So I think either (a) reframe the contribution as "CNN + jointly-trained interpretable tabular head with LLM-driven shape-function selection,"; or (b) introduce a genuine symbolic component (rule chaining, logical constraints, differentiable theorem proving, probabilistic logic) and show empirically that it contributes. The current framing overclaims relative to the architecture.

3. Ablate end-to-end joint training. Given that CNN and GAM communicate only through additive logit fusion and the CNN is pretrained in stage 1, the joint stage 2 is effectively fine-tuning. Please add a sequential-training baseline: pretrain CNN, freeze, fit GAM on residuals. If the gap to joint training is small (which the architecture suggests), the "end-to-end joint optimization" framing should be revised.

4. Need more info about the GPT-4o baseline. A general-purpose LLM at 36.02% accuracy on a binary task is anomalous and currently uninterpretable. Please specify the exact prompt, input modalities (was the MRI provided? as image, summary, or not at all?), how outputs were parsed, and whether any prompt engineering was attempted. It feels like the 4o model is not properly tuned for this task.

---

### Review · Reviewer_EY7n · 2026-05-04

**Summary Of Contributions:**

This work presents a framework for Alzheimer’s Disease (AD) Diagnosis, which combines two types of input, MRI data and text-based medical information. The framework consists in first performing an initial diagnosis of the AD using the MRI data and then refining it by means of symbolic reasoning on the medical information. The contributions are the bi-modal framework itself, an automated knowledge acquisition module for rule selection and report generation, and a set of experiments.

**Audience:**

Yes

**Audience Explanation:**

- The problem is of high relevance for both medical and machine learning communities.
- The proposed framework sound interesting, although more details must be provided for reproducibility.
- The results show promise for real escenarios.

**Broader Impact Concerns:**

Most relevant potential issues are already highlighted by the authors. My only concern would be about ensuring the consistency of the data. More concretely, often in clinical escenarios, capturing data is prone to typos, interpretations, and different way for expressing the same idea. The potential positive impact of this proposal might be compromised due to these variations. Some suggestions should be given on this regard.

**Claims And Evidence:**

No

**Claims Explanation:**

For details, see the section of requested changes:
- Several details about the data are unclear, making it difficult to see the support about the proposed framework.
- It is unclear if the single-mode models use the same set of hyperparameters than the integrated framework.
- Only one example of the rules is provided.
- There are not training details for the symbolic reasoning module.
- In medical escenarios, recall is often the crucial metric. However, the proposed framework falls in second place for this metric.
- There are not results about using only the symbolic reasoning module.

**Requested Changes:**

- Define all variables at their first occurrence. For instance, superindices m, n, r, and k in section 3.1
- Define all acronyms at their first occurrence. For instance, CN in section 3.1
- Only during the results section it becomes clear that the framework first trains the MRI module, and then jointly refines both modules. Through section 3, sometimes it reads as if it is a cascade pipeline, and some other like a joint-parallel approach. Please clarify it already in beginning of section 3.
- Please, indicate in section 3.2 that the module for MRI is open to particular architectures, which will be presented later. In its current form, it reads like there something that was forgotten.
- In machine learning, it is common to use 'y' for the ground-truth and \hat{y} for the prediction. Please, fix the notation to the standard form.
- The R_j in the first paragraph of section 3.3 has a wrong notation.
- Details about the set of rules must be provided. Perhaps in an appendix.
- In section 3.3, variables are given text-based names. This is great. But their meaning must be also explicitly given for the non-medical readers.
- There not details about the optimization method for training the parameters of the symbolic reasoning module. Please provide this information.
- Please, clarify why delta_i appear twice in Eq (4).
- Ablation studies are missing for the escenario of symbolic reasoning alone.
- Please clarify if the standard normalization mentioned in section 3.5 happens channel-wise, MRI-wise, or database-wise. Also, indicate, at least, what is the performance obtained without applying it.
- Details about the statistics of the data must be provided, including size of training, validation, and test sets; whether variables remain the same or vary across these tree sets; are the two classes (CN and AD) consistent internally in terms of their variables after been created by joining data with different labels?.
- Why did the authors decided to use a bi-variate output for a binary classification problem? Why not using a sigmoid single output? Please provide a justification for this.
- Results in Table 1 and Table 2 seem to be for the test set. That is correct. However, it would be also interesting to see the results for the training set, so we can perceive the potential for generalization or overfitting. Please include those results.

---

### Review · Reviewer_EXGL · 2026-06-05

**Summary Of Contributions:**

This works proposes a realistically clinical data dependent multimodal framework to mimic expert reasoning and diagnosis Alzheimer's Disease. With the use of symbolic reasoning, the explainability gets increased, which is important for medical field.

**Audience:**

Yes

**Audience Explanation:**

the group-wise analysis demonstrated aligns well with actual clinical findings, making it great potential toward an automatic and explainable AD diagnose system.

**Claims And Evidence:**

No

**Claims Explanation:**

The overall design looks great considering both reality and explainability. However, Recall is usually considered as a more important metric than accuracy, which is to measure how many actual positive cases are detected. From Table 1, the proposed multi-modal method shows a worse Recall score than the unimodal baseline 3D ResNet, making it less convincing. The multi-modal design has much greater potential though.

**Requested Changes:**

1. the experiment/design need to be improved to demonstrate the superiority of multi-modal design than 3D ResNet for Recall.

---

### Note · Authors · 2026-07-13

I have read and agree with the venue's withdrawal policy on behalf of myself and my co-authors.